# Effects of Melatonin on Lipid Metabolism and Circulating Irisin in Sprague-Dawley Rats with Diet-Induced Obesity

**DOI:** 10.3390/molecules25153329

**Published:** 2020-07-22

**Authors:** Yu-Tang Tung, Pei-Chin Chiang, Ya-Ling Chen, Yi-Wen Chien

**Affiliations:** 1Graduate Institute of Metabolism and Obesity Sciences, Taipei Medical University, Taipei 110, Taiwan; f91625059@tmu.edu.tw; 2Nutrition Research Center, Taipei Medical University Hospital, Taipei 110, Taiwan; 3School of Nutrition and Health Sciences, Taipei Medical University, Taipei 110, Taiwan; lucky90509@hotmail.com (P.-C.C.); ylchen01@tmu.edu.tw (Y.-L.C.); 4Research Center of Geriatric Nutrition, College of Nutrition, Taipei Medical University, Taipei 110, Taiwan

**Keywords:** melatonin, obesity, irisin, lipid metabolism, browning effect

## Abstract

Melatonin, a pivotal photoperiodic signal transducer, may work as a brown-fat inducer that regulates energy balance. Our study aimed to investigate the effects of melatonin treatment on the body fat accumulation, lipid profiles, and circulating irisin of rats with high-fat diet-induced obesity (DIO). Methods: 30 male Sprague-Dawley rats were divided into five groups and treated for 8 weeks: vehicle control (VC), positive control (PC), MEL10 (10 mg melatonin/kg body weight (BW)), MEL20 (20 mg/kg BW), and MEL50 (50 mg/kg BW). The vehicle control group was fed a control diet, and the other groups were fed a high-fat and high-calorie diet for 8 weeks to induce obesity before the melatonin treatment began. Melatonin reduced weight gain without affecting the food intake, reduced the serum total cholesterol level, enhanced the fecal cholesterol excretion, and increased the circulating irisin level. Melatonin downregulated the fibronectin type III domain containing 5 (FNDC5) and lipoprotein lipase (LPL) mRNA expressions of inguinal white adipose tissue (iWAT) and induced the browning of iWAT in both the MEL10 and MEL20 groups. Conclusion: Chronic continuous melatonin administration in drinking water reduced weight gain and the serum total cholesterol levels. Additionally, it enhanced the circulating irisin, which promoted brite/beige adipocyte recruitment together with cholesterol excretion and contributed to an anti-obesity effect.

## 1. Introduction

Globeisty refers to the worldwide increase in excess weight and obesity according to the World Health Organization (WHO). Being overweight and obese are defined as values of the body mass index (BMI), calculated as the weight in kilograms divided by the square of the height in meters, of ≥25 and 30 kg/m^2^, respectively. In light of the WHO estimates, over 1.9 billion people were overweight and, of these, over 650 million were obese in 2016—nearly 39% and 13% of the world’s adult population [1]. Obesity results from an imbalance between energy uptake and energy expenditure, and it is considered to increase the incidences of chronic diseases such as cardiovascular diseases, type 2 diabetes, and cancers [1,2].

Obesity is not only defined as a condition of abnormal fat accumulation but is also recognized as a disease that affects the quality of life and raises the risk of chronic diseases. In modern societies, multiple lifestyle characteristics, such as high-calorie food intake, sedentary behaviors, and a lack of physical exercise, are regularly reported as the causes of weight gain. An increasing number of studies have indicated that circadian misalignment may lead to obesity [3,4]. Practically, shift work, jet lag, and certain conditions that disrupt the normal dark/light, sleep/wake, and fasting/feeding cycles can be defined as circadian misalignment [5].

Melatonin (*N*-acetyl-5-methoxytryptamine), an endocrine hormone derived from tryptophan, is synthesized and secreted by the mammalian pineal gland primarily during the dark phase of the circadian cycle. Melatonin has been found in most living organisms and serves as an antioxidant against oxidative stress [6]. A number of experimental studies have shown that melatonin has chronobiotic, cytoprotective, antioxidant, anti-estrogenic, anti-inflammation, immune system modulation, and oncostatic activities [7,8,9,10,11]. BOJKOVÁ et al. [12] showed that 48 h fasting after prolonged melatonin significantly altered the carbohydrate and lipid metabolism alterations of young rats. MARKOVÁ et al. [13] pointed that melatonin administration significantly decreased the serum triacylglycerol concentration and liver glycogen content in male rats, and increased the liver phospholipid content in female rats. Nowadays, numerous studies have shown the relationship of melatonin with body weight (BW) and energy balance [14,15,16,17]. Xu et al. showed that [18] melatonin may be used as a probiotic agent to reverse high fat diet (HFD)-induced gut microbiota dysbiosis. Favero et al. [19] pointed out that melatonin supplementation ameliorates obesity-induced adipokine alteration by regulating inflammatory infiltration. Previous studies have demonstrated that a pinealectomy caused glucose intolerance in rats, and they then had a higher accumulation of adipose deposits [20,21]. Other observations indicated that nocturnal melatonin administration for 3 months to middle-aged male rats suppressed the BW gain and intra-abdominal adiposity compared to the controls [22].

In recent years, melatonin is considered to have the ability to increase the capacity of non-shivering thermogenesis through regulating brown adipose tissue (BAT) [23,24]. In light of the pressing need to check the progression of obesity, our study aimed to investigate the effects of the chronic administration of melatonin on body fat accumulation and lipid profiles. We particularly investigated irisin, a myokine associated with the browning effect of white adipose tissues (WATs), as a biomarker of brite/beige adipocytes.

## 2. Results

To examine the effects of melatonin on an obese model caused by the regular intake of calorie-dense foods, we used DIO rats as study subjects. As expected, after 8 weeks of induction the DIO rats weighed 15.5% more than the control rats. At the end of the melatonin supplementation, the body weight of the DIO rats was still significantly greater than the VC, while the total weight gains of the DIO rats treated with melatonin were significantly lower than those of PC group. However, there was no significant difference in the total weight gains between different doses of melatonin. There were no differences among groups in the relative liver, kidney, or eWAT weights (Table 1).

During the long-term use of melatonin, the food intake of the DIO rats did not significantly differ compared to the VC group; however, the rats fed the HF diet had a greater energy intake than the control rats, as shown in Table 2. The feed efficiency was higher in the DIO groups than the VC, but was significantly reduced in the melatonin-administered groups, regardless of the dose. However, there was no significant difference in the feed efficiency between the different doses of melatonin. The completion rate of all groups was around 90%, with no difference between groups, but the water amount in the MEL50 group was significantly lower than that in the VC.

To evaluate the ability of melatonin to ameliorate the lipid profile, we measured the serum, hepatic, and fecal lipid levels. Table 3 shows that both the serum TC and HDL-C levels in the PC group were higher than those in the VC group but were significantly lower in all the MEL groups when compared to the PC group. However, there was no significant difference in the serum TC and HDL-C levels between different doses of melatonin. Neither the HF diet nor the melatonin affected the hepatic fat accumulation. Figure 1 shows the level of cholesterol in the feces, and the level was remarkably higher with the HF diet than with the control diet, and that of the PC group increased by 681% compared to the VC group. Furthermore, the melatonin supplementation enhanced cholesterol excretion, especially in the MEL50 group (175% vs. the PC). This showed that melatonin can increase the cholesterol excretion.

Irisin, a myokine, is considered to be a mediator of the browning of WATs. As shown in Figure 2, the circulating irisin levels in the PC group were significantly reduced to 22.2% compared to the VC group, and the high-dose melatonin supplementation (MEL50) significantly increased the circulating irisin level compared to the PC group.

A larger adipocyte size in inguinal white adipose tissues was observed in the PC group compared with the VC group (Figure 3). After 8 weeks of the melatonin administration, MEL10, MEL20, and MEL50 significantly decreased in the large adipocytes compared with the PC group. White and brown-fat-like (brite/beige) adipocytes have distinct morphological characteristics after H&E staining. White adipocytes have a large lipid droplet full of TG and contain few mitochondria. Brite/beige adipocytes contain small, multilocular lipid droplets, and are rich in active mitochondria and cytochromes. As illustrated in Figure 3, the melatonin treatment in the MEL10 and MEL20 groups induced brite/beige adipocytes in the iWATs of DIO rats. However, brite/beige adipocytes were not discovered in MEL50.

In contrast, at doses of 20 and 50 mg/kg BW, melatonin significantly downregulated the relative expression of fibronectin type III domain containing 5 (FNDC5) by 23.6% and 15.9% (vs. the PC group), respectively. To assess the browning effect of melatonin on iWATs, we examined the related markers, including peroxisome proliferator-activated receptor gamma coactivator 1α (PGC-1α) and uncoupling protein 1 (UCP1). The results showed that melatonin did not affect the expression of PGC-1α or UCP1 (Figure 4).

Additionally, we investigated the biomarkers related to lipid metabolism (Figure 5). The lipoprotein lipase (LPL) mRNA expressions in the MEL20 and MEL50 groups were downregulated by 92.5%, 27.2%, and 64.1% (vs. the PC group), respectively. However, there was no significant difference in the LPL mRNA expression between different doses of melatonin. There was no remarkable observation of the gene expression of hormone-sensitive lipase (HSL), adiponectin, or peroxisome proliferator-activated receptor γ (PPARγ).

## 3. Discussion

A previous study demonstrated that rats orally administered melatonin by gavage were averse to doses of ≥50 mg/kg BW [25]. Long-term melatonin administration at a lower dose halted weight gain and decreased the visceral fat accumulation in middle-aged rats [22,26]. Rats fed the HF diets and treated with 5 or 10 mg melatonin/kg BW for 48 weeks had lower total BWs [27], and 30 mg of melatonin for 12 weeks also decreased the BW but did not affect the mass of eWATs [15]. Raskind et at. [28] demonstrated that melatonin supplementation decreased the retroperitoneal, omental, and mesenteric WATs; in contrast, Wolden-Hanson et al. indicated that melatonin treatment decreased eWATs but did not affect the masses of the retroperitoneal, omental, or mesenteric WATs [26]. This study demonstrated that chronic continuous melatonin administration (at least 10 mg/kg BW) in drinking water reduced the total weight gain and feed efficiency, but not the eWAT mass.

Irisin, a myokine cleaved from FNDC5 in muscle and secreted into the circulatory system, has the ability to facilitate the UCP1 expression in WATs and make WATs act as BATs through non-shivering thermogenesis [29]. Bostrom et al. [29] pointed out irisin enhanced energy expenditure and ameliorated weight gain and glucose tolerance. However, to date there have been inconsistent results as to the association of irisin with BMI. Some studies have reported that circulating irisin levels were positively correlated with the BMI [30,31], while others showed a negative correlation [32,33]. Jimenez-Aranda et al. indicated that lean rats had higher circulating irisin levels than obese rats, but melatonin treatment induced the appearance of brite/beige adipocytes in iWATs but did not significantly affect the irisin levels [33]. In the present study, a larger adipocyte size in inguinal white adipose tissues was observed in the PC group compared with the VC group. However, long-term melatonin administration significantly decreased in the large adipocytes compared with the PC group. Additionally, 10 and 20 mg/kg of melatonin induced brite/beige adipocytes in iWATs. A previous study [34] showed that 10 mg/kg of melatonin can improve the mitochondrial function of the iWATs in Zucker diabetic obese rats. Melatonin affects the mitochondrial respiratory control rate caused by beige fat and WAT [34]. In addition, melatonin reduces the mitochondrial oxidation state by increasing the superoxide dismutase activity and reducing nitrite [34]. In addition, high-dose melatonin supplementation (MEL50) significantly enhanced the circulating irisin level compared to the PC group. FNDC 5/irisin has recently been identified as a novel protein that can stimulate the browning of white fat by inducing thermogenesis through increasing UCP1 [35]. In the study, we found that melatonin could increase circulating irisin level, which binds to the surface of white adipocytes to induce the expression of UCP1 and trigger the transformation of white adipocytes into brown adipocytes. The browning of adipose tissue causes the dissipation of energy in the form of heat without ATP formation, which may enhance fatty acid oxidation and increase energy expenditure through non-shivering thermogenesis. Therefore, melatonin inhibites the occurrence and development of obesity through enhancing the circulating irisin level. Long-term melatonin administration also significantly downregulated FNDC5 mRNA and slightly decreased UCP1 mRNA to restore the irisin resistance status of iWATs. Therefore, the melatonin increase in circulating irisin, resulting from the cleavage of the membrane protein FNDC5, was shown to induce adipocyte browning, with increased lipid oxidation and thermogenesis. In the study, we found that melatonin induced WAT browning, which is a promising means to increase energy expenditure and improve lipid metabolism. In the current study, we were unable to detect a significant increase in PGC1α expression in melatonin-treated groups. The result is similar to a previous study [36] that shows an additional regulation of FNDC5 independent of PGC1α in adipose tissue.

Additionally, melatonin significantly decreased the TC level and increased the HDL-C level compared to the PC group. The hypolipidemic effect of melatonin in our study may have resulted from the increased cholesterol excretion in the feces. A previous study showed that fecal cholesterol excretion may be increased by stimulating bile cholesterol secretion and reducing intestinal cholesterol absorption [37]. A previous study observed that melatonin significantly reduced the cholesterol absorption in rats fed a high-cholesterol diet and ameliorated plasma lipid profiles [38]. Moreover, numerous studies indicated that the hypocholesterolemic effect of melatonin was due to endogenous cholesterol clearance mechanisms via bile acid synthesis and the inhibition of LDL receptor activity, but not the alteration of fatty acid synthesis [39,40]. Therefore, in this study we found that melatonin reduced body weight gain in two ways, including increasing circulating irisin levels and enhancing fecal cholesterol excretion.

## 4. Materials and Methods

### 4.1. Reagents

All the reagents were obtained from commercial suppliers and were of analytical grade. The melatonin was purchased from Sigma-Aldrich (St. Louis, MO, USA).

### 4.2. Animals and Experimental Protocols

Thirty male Sprague-Dawley rats were purchased from BioLASCO Taiwan (Taipei, Taiwan) at 7 weeks of age and housed 2 per cage. All the plastic cages were transparent with wire tops. Standard conditions were maintained, with a controlled temperature (25 ± 2 °C) and humidity (65% ± 5%) and a 12 h light/dark cycle. The rats were fed Rodent Lab Chow 5001 (PMI, St. Louis, MO) for 1 week during acclimatization. During an 8-week obesity-induction period, 24 rats were fed a high-fat (HF) diet containing 23.5% soybean oil (comprising 45% of the energy), while the other 6 rats were fed a control diet with 7% soybean oil (comprising 16% of the energy). The two diets were based on the American Institute of Nutrition-93M (AIN-93M) diet (Table 4).

After the induction period, the rats were divided into five groups and treated for 8 weeks; the groups were vehicle control (VC), positive control (PC), MEL10 (10 mg melatonin/kg BW), MEL20 (20 mg/kg BW), and MEL50 (50 mg/kg BW). The VC group was fed a control diet, and the other groups were diet-induced obese (DIO) rats fed HF diets for 8 weeks before the melatonin treatment. During the chronic treatment period, the rats continued to be fed the same diets as before, and the melatonin treatment was added to the drinking water. In addition, to prevent the photodegradation of the melatonin, the water bottles were covered with aluminum foil. Replacing the water and adjusting the volume every 2–3 d not only prevented melatonin degradation but promoted dose achievement through the route of continuous administration. The melatonin was dissolved in a minimum volume of absolute ethanol and then added to the drinking water; the ethanol dose equaled 1 mL/kg BW/day in each group to eliminate its influence. Fresh melatonin and vehicle solutions were prepared three times a week, and the melatonin and ethanol doses were adjusted to the BW weekly. The completion rate of melatonin treatment was calculated as the recorded achieved dose/target dose × 100%. Feces were directly collected by softly touching the area surrounding the anus to stimulate the defecation reflex at week 7 of the chronic administration.

During the study period, food and water were supplied ad libitum, and the amounts were recorded to evaluate the feed efficiency (%) and actual melatonin dose. At the end of the chronic treatment period, the rats were sacrificed after being starved overnight for 12 h. Blood was withdrawn from the abdominal aorta under anesthesia (an intramuscular injection of 1 mL/kg BW of a Zoleti-Rompun mixture) and centrifuged at 2350× *g* for 10 min at 4 °C to separate the serum. The unguinal WAT (iWAT), epididymal WAT (iWAT), and the liver were immediately separated and washed with normal saline, and the samples were stored at −80 °C until analysis. All the procedures were approved by the Institutional Animal Care and Use Committee of Taipei Medical University (LAC-2014-0345).

### 4.3. Serum, Hepatic, and Fecal Lipid Measurements

The serum triglyceride (TG), total cholesterol (TC), high-density lipoprotein cholesterol (HDL-C), and low-density lipoprotein cholesterol (LDL-C) levels were determined with the ADVIA 1800 Chemistry System (Siemens, Tarrytown, NY, USA) using accordant reagents. The hepatic and fecal lipid extractions were carried out following the method of Folch et al. [41]. The hepatic TC and TG and fecal cholesterol were analyzed using commercial diagnostic reagents (Randox Laboratories, Ltd. (Crumlin, Co., Antrim, UK).

### 4.4. Real-Time Reverse-Transcription Polymerase Chain Reaction (RT-PCR)

The total RNA was extracted from the liver tissue using the TRI-Reagent (Sigma-Aldrich, Steinheim, Germany), and the iWAT was extracted with a GENEzol TriRNA Pure Kit (Geneaid Biotech, New Taipei City, Taiwan), both according to the manufacturer’s instructions, and 3 µg of total RNA was reverse transcribed with a RevertAid First Strand cDNA Synthesis Kit (Thermo Scientific, Waltham, MA, USA). A quantitative real-time PCR was performed according to the SYBR Green/ROX qPCR Master Mix (2X) (Thermo Scientific, Waltham, MA, USA) in a 25 μL total reaction volume using the 7300 Real-Time PCR System (Applied Biosystems, Foster City, CA, USA). The primers used in this study are presented in Table 5. The mRNA expression was normalized against the GAPDH gene as a control and expressed as a multiple of change relative to the control.

### 4.5. Serum Irisin Levels

An irisin enzyme linked-immunosorbent assay (ELISA) kit (BioVendor, Brno, Czech Republic) was used to determine the irisin concentrations in the serum, following the manufacturer’s instructions. The sensitivity was 1 ng/mL, the intra-assay coefficient of variation was 6–8%, and the interassay precision was 8–10%.

### 4.6. Histologic Analysis

The iWATs were fixed in 10% buffered formaldehyde. Before proceeding to the next step, the tissues were soaked in absolute ethanol for one night and embedded in paraffin. The sections were then stained with hematoxylin and eosin (H&E).

### 4.7. Statistical Analysis

The data were expressed as the mean ± standard error of the mean (SEM). Differences between the groups were analyzed by a one-way analysis of variance (ANOVA), followed by Duncan’s multiple-range test and Dunnett’s T3 test as a post-hoc test with PASW Statistics for Windows, vers. 18.0 (Chicago, IL, USA). Statistical significance was considered at *p* < 0.05.

## 5. Conclusions

The study had the limitation that melatonin was added to tap water and available ad libitum, and rats in the MEL50 group spontaneously reduced their water consumption due to a dislike for the taste. In summary, chronic continuous melatonin administration in drinking water reduced weight gain and serum total cholesterol levels. Additionally, it enhanced circulating irisin, thus promoting brite/beige adipocyte recruitment together with cholesterol excretion, and contributed to an anti-obesity effect. Melatonin, thanks to this effect, can play an active role in the clinical course of obesity. However, our study also had some limitations, including only using an animal study, and the dose of melatonin to provide to humans is not clear. These limitations attenuate our ability to generalize conclusions about the effects of melatonin on anti-obesity. Therefore, further research and clinical trials are needed to evaluate the efficacy and safety of melatonin for humans—melatonin is still a hormone.

## Figures and Tables

**Figure 1 molecules-25-03329-f001:**
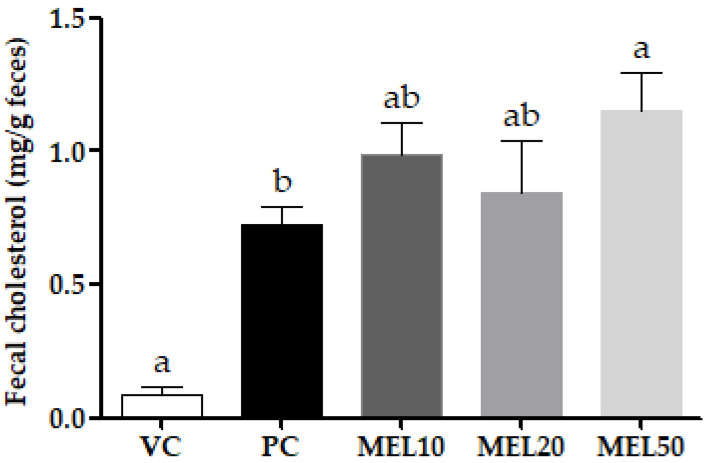
Effect of the chronic administration of melatonin on the fecal cholesterol level. Values are the mean ± SEM (*n* = 5 or 6). ^a,b^
*p* < 0.05 by Duncan’s multiple-range test. VC, vehicle control; PC, positive control; MEL10, MEL20, and MEL50 are 10, 20, and 50 mg melatonin/kg body weight, respectively.

**Figure 2 molecules-25-03329-f002:**
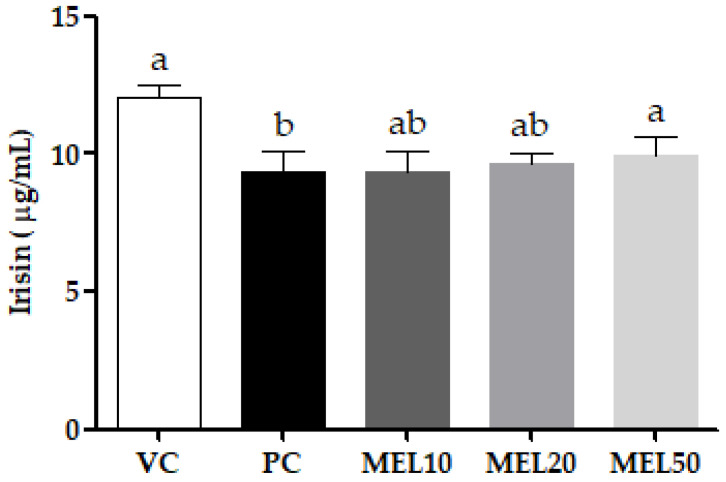
Effects of the chronic administration of melatonin on the circulating irisin levels. Values are the mean ± SEM (*n* = 4–6). ^a,b^
*p* < 0.05 by Duncan’s multiple-range test.

**Figure 3 molecules-25-03329-f003:**
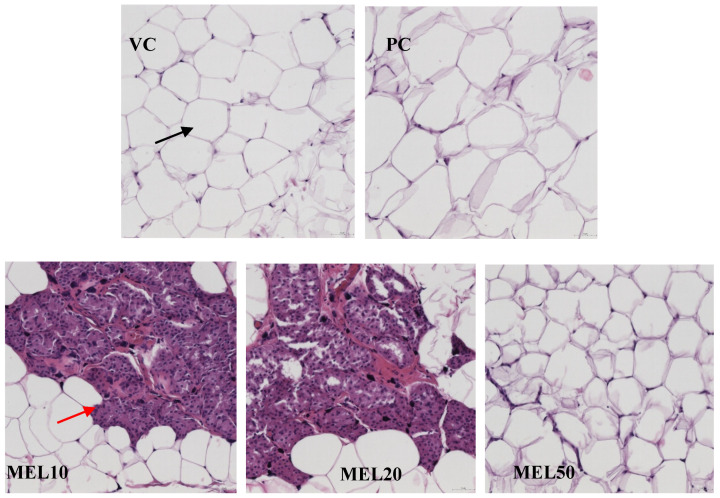
Effects of melatonin on the adipocyte morphology in inguinal white adipose tissues (hematoxylin and eosin (H&E) stain, x20). Black arrowhead indicates white adipose tissue; red arrowhead indicates beige adipose tissue.

**Figure 4 molecules-25-03329-f004:**
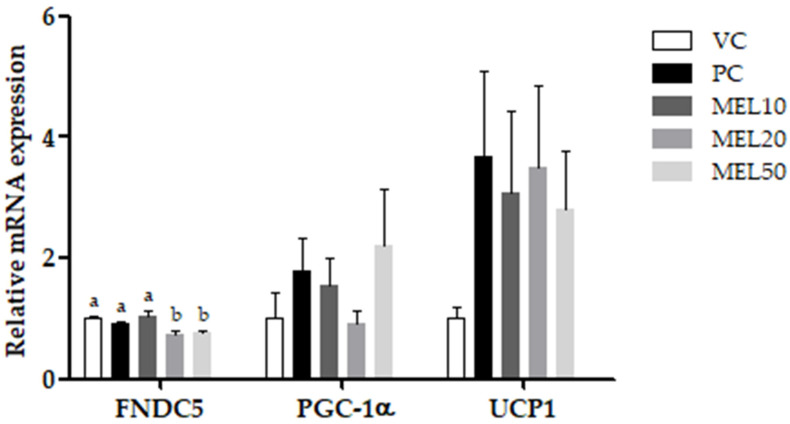
Effects of the chronic administration of melatonin on the relative mRNA expressions of fibronectin type III domain containing 5 (FNDC5), peroxisome proliferator-activated receptor gamma coactivator 1α (PGC-1α), and uncoupling protein 1 (UCP1) in inguinal white adipose tissue. Values are the mean ± SEM (*n* = 6). ^a,b^
*p* < 0.05 by Duncan’s multiple-range test.

**Figure 5 molecules-25-03329-f005:**
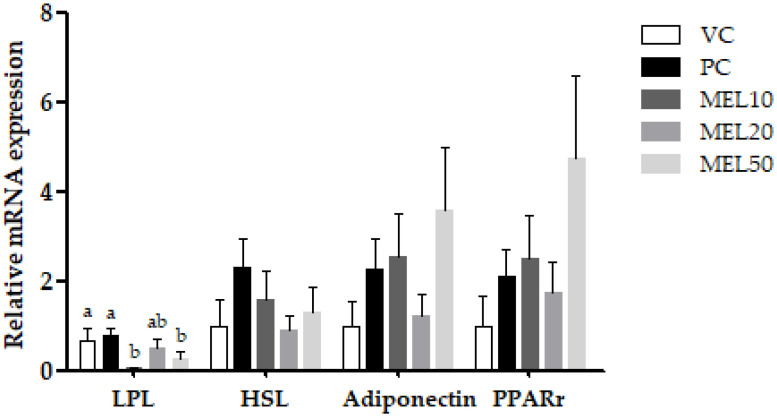
Effects of the chronic administration of melatonin on the relative mRNA expressions of lipoprotein lipase (LPL), hormone-sensitive lipase (HSL), adiponectin, and peroxisome proliferator-activated receptor γ (PPARγ) in inguinal white adipose tissues. Values are the mean ± SEM (*n* = 46). ^a,b^
*p* < 0.05 by Duncan’s multiple range test.

**Table 1 molecules-25-03329-t001:** Body, body fat, and selected organ weights after the chronic administration of melatonin for 8 weeks.

	VC			PC			MEL10			MEL20			MEL50		
Initial body weight (g)	535	±	12 ^b^	635	±	12 ^a^	603	±	18 ^a^	614	±	18 ^a^	619	±	23 ^a^
Final body weight (g)	597	±	12 ^b^	745	±	18 ^a^	688	±	22 ^a^	699	±	25 ^a^	704	±	32 ^a^
Total weight gain (g)	62.3	±	5.5^c^	110.2	±	6.0 ^a^	84.8	±	6.2 ^b^	85.5	±	8.3 ^b^	84.8	±	9.4 ^b^
Liver weight (g)	16.5	±	0.5	18.5	±	1.4	16.8	±	0.6	17.5	±	0.8	17.5	±	1.4
Kidney weight (g)	3.75	±	0.19	4.37	±	0.25	3.65	±	0.21	3.81	±	0.32	4.04	±	0.22
eWAT weight (g)	13.0	±	0.8 ^b^	22.4	±	1.4 ^a^	20.7	±	2.1 ^a^	16.8	±	2.0 ^a,b^	19.9	±	2.5 ^a^
Relative liver weight (g/100 g BW)	2.77	±	0.12	2.47	±	0.13	2.44	±	0.07	2.50	±	0.06	2.47	±	0.10
Relative kidney weight (g/100 g BW)	0.63	±	0.03	0.59	±	0.03	0.53	±	0.03	0.55	±	0.05	0.58	±	0.05
Relative eWAT weight (g/100 g BW)	2.18	±	0.13	3.03	±	0.23	3.00	±	0.27	2.40	±	0.29	2.79	±	0.28

Values are the mean ± SEM (*n* = 6). ^a,b,c^ Values with different superscripts significantly differ (*p* < 0.05). eWAT, epididymal white adipose tissue; BW, body weight; VC, vehicle control; PC, positive control; MEL10, MEL20, and MEL50, are 10, 20, and 50 mg melatonin/kg body weight, respectively.

**Table 2 molecules-25-03329-t002:** Diet and water intake, feed efficiency, and completion rate during the long-term use of melatonin.

	VC			PC			MEL10			MEL20			MEL50		
Food intake (g/d)	21.4	±	0.5	20.7	±	0.5	19.8	±	0.4	20.6	±	0.4	20.3	±	0.6
Dietary caloric intake (kcal/d)	84.3	±	1.8 ^b^	97.2	±	2.3 ^a^	93.0	±	2.0 ^a^	97.0	±	1.7 ^a^	95.2	±	2.9 ^a^
Feed efficiency (%)	5.19	±	0.40 ^c^	9.52	±	0.49 ^a^	7.67	±	0.55 ^b^	7.42	±	0.72 ^b^	7.50	±	0.87 ^b^
Water intake (mL/d)	29.6	±	0.7 ^a^	28.8	±	2.4 ^a,b^	25.9	±	0.9 ^a,b^	31.1	±	2.9 ^a,b^	22.4	±	0.5 ^b^
Completion rate (%)	90.9	±	1.1	91.1	±	1.6	89.7	±	1.6	91.7	±	1.1	91.3	±	0.5
Actual melatonin dose (mg/kg BW)	0.0	±	0.0	0.0	±	0.0	9.0	±	0.2 ^c^	18.2	±	0.3 ^b^	45.6	±	0.2 ^a^

Values are the mean ± SEM (*n* = 6). ^a,b,c^ Values with different superscripts significantly differ (*p* < 0.05). Formulas: feed efficiency = weight gain/food intake × 100%; completion rate = achieved dose/target dose × 100%. BW, body weight; VC, vehicle control; PC, positive control; MEL10, MEL20, and MEL50 are 10, 20, and 50 mg melatonin/kg body weight, respectively.

**Table 3 molecules-25-03329-t003:** Lipid profiles in the serum and liver tissues after the chronic administration of melatonin for 8 weeks.

	VC			PC			MEL10			MEL20			MEL50		
TG (mg/dL)	51.2	±	5.8	43.2	±	4.9	36.7	±	2.7	32.7	±	3.9	38.7	±	6.4
TC (mg/dL)	59.7	±	5.6 ^b^	86.3	±	7.1 ^a^	67.3	±	5.3 ^b^	64.3	±	4.2 ^b^	62.0	±	8.5 ^b^
HDL-C (mg/dL)	10.3	±	0.8 ^c^	19.0	±	1.1 ^a^	14.5	±	1.4 ^b^	14.3	±	0.8 ^b^	12.8	±	1.5 ^b,c^
LDL-C (mg/dL)	5.00	±	0.52	5.50	±	0.50	4.83	±	0.48	4.67	±	0.33	4.00	±	0.37
Hepatic TC (mg/g liver)	2.46	±	0.15	2.94	±	0.36	2.98	±	0.07	2.71	±	0.27	2.64	±	0.15
Hepatic TG (mg/g liver)	11.2	±	1.6	16.0	±	2.0	14.2	±	1.1	11.2	±	0.6	12.7	±	1.6

Values are the mean ± SEM (*n* = 6). ^a,b,c^ Values with different superscripts significantly differ (*p* < 0.05). TG, triglyceride; TC, total cholesterol; HDL-C, high-density lipoprotein cholesterol; LDL-C, low-density lipoprotein cholesterol. VC, vehicle control; PC, positive control; MEL10, MEL20, and MEL50 are 10, 20, and 50 mg melatonin/kg body weight, respectively.

**Table 4 molecules-25-03329-t004:** Ingredients of the experimental diets.

g/kg Diet	Control Diet	High-Fat and -Calorie Diet
Corn starch	529.49	289.23
Casein	200	238.12
Sucrose	100	119.08
Cellulose	50	59.54
Soybean oil	70	235.07
AIN 93M mineral mix	35	41.68
AIN 93M vitamin mix	10	11.91
Choline bitartrate	2.5	2.98
L-cystine	3	2.38
Tertbutylhydroquinone	0.014	0.017
Calories (kcal/kg diet)	3947.96	4701.35

**Table 5 molecules-25-03329-t005:** Primer sequences for the real-time polymerase chain reaction.

Gene (Accession No.)		Sequence (5′→3′)
LPL (NM_012598.2)	Fw	GATGGACGGTGACAGGAATGTA
	Rv	CGGCAGACACTGGATAATGTTG
HSL (XM_008758896.1)	Fw	GCTGGGCTGTCAAGCACTGT
	Rv	GTAACTGGGTAGGCTGCCAT
PPARγ (XM_006237009.2)	Fw	GCCCTTTGGTGACTTTATGGAG
	Rv	GCAGCAGGTTGTCTTGGATGT
Adiponectin (NM_144744.3)	Fw	CGTTCTCTTCACCTACGACCAGT
	Rv	ATTGTTGTCCCCTTCCCCATAC
UCP1 (NM_012682.2)	Fw	AGAAGGATTGCCGAAACTGTAC
	Rv	AGATCTTGCTTCCCAAAGAGG
PGC-1α (XM_008770220.1)	Fw	ACCAAACCCACAGAGAACAG
	Rv	GGGTCAGAGGAAGAGATAAAGTTG
FNDC5 (NM_001270981.1)	Fw	AGGACAACGAGCCCAATAAC
	Rv	CATATCTTGCTTCGGAGGAGAC
GAPDH (NG_028301.2)	Fw	ACAGCAACAGGGTGGTGGAC
	Rv	TTTGAGGGTGCAGCGAACTT

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
