# Peer review of "Effects of Melatonin on Lipid Metabolism and Circulating Irisin in Sprague-Dawley Rats with Diet-Induced Obesity"

_molecules, 2020, doi:10.3390/molecules25153329_

Round 1

Reviewer 1 Report

I believe that the manuscript lack of several citations of my own grop and others, working in zucker rats that came out in the earlies 2000 until 2012. There is an inconstance of the results when comparing doses.

It is difficult to understand some of the results considering de initial weights.

Also de significant differences should be better explained to achieve the real significant differences.

It is difficult to understand the with some doses of melatonin, there is a reduction in body weight but not in the fatty deposists.

I think the a review of the manuscript is necessary before its acceptance

Author Response

Q1. I believe that the manuscript lack of several citations of my own grop and others, working in zucker rats that came out in the earlies 2000 until 2012.

Answer: Thanks for reviewer suggestion. I added several recently published citations (Page 2, Lines 50-60).

Q2. There is an inconstance of the results when comparing doses.

Answer: Thanks for reviewer suggestion. I have compared the different doses and modified them in the results (Page 5, Lines 146-147, 157-158; Page 6, Lines 176-177; Page 8, Lines 220-221).

Q3. It is difficult to understand some of the results considering de initial weights.

Answer: The VC group was fed a control diet, and the other groups were diet-induced obese (DIO) rats fed HF diets for 8 weeks before melatonin treatment. As expected, after 8 weeks of induction, DIO rats weighed 15.5% heavier than control rats. In this study, we studied the therapeutic effect of melatonin rather than the protective effect. Therefore, the initial weights of DIO rats should be higher than the normal rats (Page 3, Lines 87-88; Page 4, Lines 143-144).

Q3. Also de significant differences should be better explained to achieve the real significant differences.

Answer: Thanks for reviewer suggestion. I have explained them in the discussion (Page 9, Lines 237-239, 248-251, 252-254, 255-257, 260-262).

Q4. It is difficult to understand the with some doses of melatonin, there is a reduction in body weight but not in the fatty deposists.

Answer: Actually, the body weight of DIO rats was significantly greater than VC, while the total weight gains of DIO rats treated with melatonin was significantly lower than that of PC group. But melatonin was only slightly decreased the relative eWAT weight. In addition, the melatonin intervention significantly decreased in the large adipocytes compared with the PC group after H&E staining.

Q5. I think the a review of the manuscript is necessary before its acceptance

Answer: Thanks for reviewer suggestion. I have reviewed them in the discussion.

Reviewer 2 Report

Manuscript entitled “Effects of Melatonin on Lipid Metabolism and Circulating Irisin in Sprague-Dawley Rats with Diet-induced Obesity” is written by experienced scientists. It meets the scope and objectives of the journal. Manuscript provides well-designed study and gives the scientific novelty of the research. Before publication, several comments must be implemented:

Introduction:

Line 49, sentence “Melatonin has been found in most living organisms and serves as an antioxidant against oxidative stress” is absolutely insufficient. Melatonin is very poorly introduced in this study. I suggest authors to emphasize the pleiotropic effects of MEL in organism. In a few sentences (2-3) please describe that MEL demonstrates chronobiotic1, cytoprotective/antioxidant2, anti-estrogenic3, anti-inflammation/immune system modulation4, and well-described oncostatic5 activities. I suggest to use these excellent references for this purpose:

1 Mol Cell Endocrinol. 2020 Mar 1;503:110687.

2 Curr Cancer Drug Targets. 2010 May;10(3):279-86.

3 Folia Biol (Praha). 2001;47(1):5-10.

4 J Pineal Res. 2018 Nov;65(4):e12525.

5 Crit Rev Oncol Hematol. 2018 Feb;122:133-143.

Lines 50-52, very important sentence “Nowadays, numerous studies….relationship of MEL with body weight and energy balance” is without references!

I strongly suggest to cite these pioneering papers on this topic:

  • Acta Vet Brno. 2006; 75(1): 21-32. https://actavet.vfu.cz/75/1/0021/
  • Acta Vet Brno 2003; 72:163-73. https://actavet.vfu.cz/72/2/0163/

Methods:

Line 71, please specify “other rats” and describe number of these animals, n=?

Results:

Line 163, please insert  ”…but were significantly lower in all MEL groups when compared to PC group”.

Lines 168-174, the sentences are repeating to sentences from lines 161-167,

Line 181, authors analyzed four DIO groups, but mentioned only one percentage (28.5) compared to VC group. Please correct.

Line 182, please insert…circulating irisin levels compared to PC group.

Figure 3. How authors explain, why brite/beige adipose tissue was not present in MEL50 group? Please discuss this result in Discussion.

Discussion:

Shortly discuss why PGC-1a and UCP1 (as markers of adipose tissue browning) did nod correlate with FNDC5 changes.

Conclusions:

Please shortly mention limitations of this study. The aim of the preclinical studies and this study must be the translation of results to clinical practice, therefore, add to Conclusions own findings addressed to clinicians.

Author Response

Q1. Line 49, sentence “Melatonin has been found in most living organisms and serves as an antioxidant against oxidative stress” is absolutely insufficient. Melatonin is very poorly introduced in this study. I suggest authors to emphasize the pleiotropic effects of MEL in organism. In a few sentences (2-3) please describe that MEL demonstrates chronobiotic, cytoprotective/antioxidant, anti-estrogenic, anti-inflammation/immune system modulation, and well-described oncostatic activities. I suggest to use these excellent references for this purpose:

1 Mol Cell Endocrinol. 2020 Mar 1;503:110687.

2 Curr Cancer Drug Targets. 2010 May;10(3):279-86.

3 Folia Biol (Praha). 2001;47(1):5-10.

4 J Pineal Res. 2018 Nov;65(4):e12525.

Answer: Thanks for reviewer suggestion. I added theses sentences and references (Page 2, Lines 50-52).

Reference

Gorman, M.R. Temporal organization of pineal melatonin signaling in mammals. Mol. Cell. Endocrinol. 2019, 503, 110687.

González, A.; Alvarez-García, V.; Martínez-Campa, C.; Mediavilla, M.D.; Alonso-González, C.; Sánchez-Barceló, E.J.; Cos, S. In vivo inhibition of the estrogen sulfatase enzyme and growth of DMBA-induced mammary tumors by melatonin. Curr. Cancer Drug Targets 2010, 10, 279-286.

Kubatka, P.; Bojková, B.; ciková-Kalická, K.M.; Mníchová-Chamilová, M.; Adámeková, E.; Ahlers, I.; Ahlersová, E.; Cermáková, M. Effects of tamoxifen and melatonin on mammary gland cancer induced by N-methyl-N-nitrosourea and by 7,12-dimethylbenz(a)anthracene, respectively, in female Sprague-Dawley rats. Folia Biol. (Praha). 2001, 47, 5-10.

Hardeland, R. Melatonin and inflammation-story of a double-edged blade. J. Pineal. Res. 2018, 65, e12525.

Q2. Lines 50-52, very important sentence “Nowadays, numerous studies….relationship of MEL with body weight and energy balance” is without references!

Answer: Thanks for reviewer suggestion. I added the references (Page 2, Line 57).

Reference

Tan, D.X.; Manchester, L.C.; Fuentes-Broto, L.; Paredes, S.D.; Reiter. R.J.; Significance and application of melatonin in the regulation of brown adipose tissue metabolism: Relation to human obesity. Obes. Rev. 2011, 12, 167-188.

Prunet-Marcassus, B.; Desbazeille, M.; Bros, A.; Louche, K.; Delagrange, P.; Renard, P.; Casteilla, L.; Pénicaud, L. Melatonin reduces body weight gain in Sprague Dawley rats with diet-induced obesity. Endocrinology 2003, 144, 5347-5352.

Sartori, C.; Dessen, P.; Mathieu, C.; Monney, A.; Bloch, J.; Nicod, P.; Scherrer, U.; Duplain, H. Melatonin improves glucose homeostasis and endothelial vascular function in high-fat diet-fed insulin-resistant mice. Endocrinology 2009, 150, 5311-5317.

Sun, H.; Wang, X.; Chen, J.; Song, K.; Gusdon, A.M.; Li, L.; Bu, L.; Qu, S. Melatonin improves non-alcoholic fatty liver disease via MAPK-JNK/P38 signaling in high-fat-diet-induced obese mice. Lipids Health Dis. 2016, 15, 202.

Q3. I strongly suggest to cite these pioneering papers on this topic:

Acta Vet Brno. 2006; 75(1): 21-32. https://actavet.vfu.cz/75/1/0021/

Acta Vet Brno 2003; 72:163-73. https://actavet.vfu.cz/72/2/0163/

Answer: Thanks for reviewer suggestion. I added theses sentences and references (Page 2, Lines 52-56).

Reference

Bojková, B.; Marková, M.; Ahlersová, E.; Ahlers, I.; Adámeková, E.; Kubatka, P.; Kassayová, M. Metabolic effects of prolonged melatonin administration and short-term fasting in laboratory rats. Acta Vet. Brno 2006, 75, 21-32.

Marková, M.; Adámeková, E.; Kubatka, P.; Bojková, B.; Ahlersová, E.; Ahlers, I. Effect of prolonged melatonin application on metabolic parameters and organ weights in young male and female Sprague-Dawley rats. Acta Vet. Brno 2003, 72, 163-173.

Q4. Methods: Line 71, please specify “other rats” and describe number of these animals, n=?.

Answer: Thanks for reviewer suggestion. I have added it in the revised manuscript (Page 2, Line 80).

Q5. Line 163, please insert ”…but were significantly lower in all MEL groups when compared to PC group”.

Answer: Thanks for reviewer suggestion. I have added it in the revised manuscript (Page 6, Lines 175-176).

Q6. Lines 168-174, the sentences are repeating to sentences from lines 161-167.

Answer: Thanks for reviewer suggestion. I have deleted it in the revised manuscript.

Q7. Line 181, authors analyzed four DIO groups, but mentioned only one percentage (28.5) compared to VC group. Please correct.

Answer: Thanks for reviewer suggestion. I have modified the sentence in the revised manuscript (Page 6, Line 187).

Q8. Line 182, please insert…circulating irisin levels compared to PC group.

Answer: Thanks for reviewer suggestion. I have deleted it in the revised manuscript (Page 7, Line 189).

Q9. Figure 3. How authors explain, why brite/beige adipose tissue was not present in MEL50 group? Please discuss this result in Discussion.

Answer: Thanks for reviewer suggestion. I have discussed it in the revised manuscript (Page 9, Lines 251-252).

Q10. Shortly discuss why PGC-1a and UCP1 (as markers of adipose tissue browning) did nod correlate with FNDC5 changes.

Answer: Thanks for reviewer suggestion. I have discussed them in the revised manuscript (Page 9, Lines 254-259).

Q11. Please shortly mention limitations of this study. The aim of the preclinical studies and this study must be the translation of results to clinical practice, therefore, add to Conclusions own findings addressed to clinicians.

Answer: Thanks for reviewer suggestion. I have added them in the revised manuscript (Page 9, Lines 268-269; Pages 9-10, Lines 273-275).

Reviewer 3 Report

Effects of Melatonin on Lipid Metabolism and Circulating Irisin in Sprague-Dawley Rats with Diet-induced Obesity

The abstract of the work is clear and concise, summarizing and synthesizing the purpose of the study. In the same way, the keywords chosen can clearly identify the theme that is then addressed.

The introduction is rich in information, but especially in the central part there is a lack of fluidity in the presentation of the concepts. It should be reviewed trying to better connect the different points exposed to each other in order to better define the relationship between them and to argue more deeply the concepts expressed.

The materials and methods are expressed clearly and synthetically. The schematic exposure in this case makes the methods comprehensible and replicable.

Tables that give a clear view of the proposed data and the methods used are also appreciated.

The first part of the Results section is well organized: the accurate description of the background (food, water) and of the characteristics of the individual animals at the end of the interventions (weight, BMI, weight of different organs) makes it easier to understand the other data presented later.

There are only some mistakes probably typing errors/views to fix (for example, the repetition of the same sentence on page 5, lines 161-174).

About the morphological part, on the contrary, it turns out to be rather deficient in some points.

The results obtained are not clearly described and this gas is also reflected in the discussion.

In the results part, it is necessary to work on the text, with a remodulation and a better explanation of what has been achieved and demonstrated with the experiments.

It would be useful that also in the figures some additions were made. It is recommended to insert some markers that better highlight the salient points of the different images.

Overall, Authors proposed beautiful, meaningful and interesting results, using methods that are not only well defined, but also have a not inconsiderable scientific value. In the discussion, unfortunately, all these points don’t find a correspondence: all the data are not very well argued and, in this way, they lose importance.

Moreover, in the Discussion section, the Authors focused their attention on some criticisms about melatonin dosage. It is an interesting point, but, also in this case, not well delineated and the point is not understandable.

Another very deficient point is the bibliography, which is not adequately updated and that is very thin.

Overall, in mine opinion, there are important outcomes obtained through these analyses, but they are not well expressed: this makes the work unclear and the communication of what has been achieved ineffective.

Already in the introduction, although the purpose of the work is clear, the reasons for supposing that melatonin can be used as a therapeutic intervention are not clear.

In the same way, at the end, the Authors draw conclusions, correct, but without having adequately discussed the results obtained and without having first gutted the mechanisms on which melatonin can work.

Results have been listed but not explained, which makes the results themselves not very useful.

Author Response

Q1. The abstract of the work is clear and concise, summarizing and synthesizing the purpose of the study. In the same way, the keywords chosen can clearly identify the theme that is then addressed.

Answer: Thanks for reviewer appreciation.

Q2. The introduction is rich in information, but especially in the central part there is a lack of fluidity in the presentation of the concepts. It should be reviewed trying to better connect the different points exposed to each other in order to better define the relationship between them and to argue more deeply the concepts expressed.

Answer: Thanks for reviewer suggestion. I added theses sentences and references (Page 2, Lines 50-60).

Q3. The materials and methods are expressed clearly and synthetically. The schematic exposure in this case makes the methods comprehensible and replicable.

Answer: Thanks for reviewer appreciation.

Q4. Tables that give a clear view of the proposed data and the methods used are also

appreciated.

Answer: Thanks for reviewer appreciation.

Q5. The first part of the Results section is well organized: the accurate description of the background (food, water) and of the characteristics of the individual animals at the end of the interventions (weight, BMI, weight of different organs) makes it easier to understand the other data presented later.

Answer: Thanks for reviewer appreciation.

Q6. There are only some mistakes probably typing errors/views to fix (for example, the repetition of the same sentence on page 5, lines 161-174).

Answer: Thanks for reviewer suggestion. I have deleted it in the revised manuscript.

Q7. About the morphological part, on the contrary, it turns out to be rather deficient in some points. The results obtained are not clearly described and this gas is also reflected in the discussion.

Answer: Thanks for reviewer suggestion. I have added the results and discussion in the revised manuscript (Page 7, Lines 194-196, 201-202; Page 9, Lines 248-252).

Q8. In the results part, it is necessary to work on the text, with a remodulation and a better explanation of what has been achieved and demonstrated with the experiments.

Answer: Thanks for reviewer suggestion. I have added the results in the revised manuscript.

Q9. It would be useful that also in the figures some additions were made. It is recommended to insert some markers that better highlight the salient points of the different images.

Answer: Thanks for reviewer suggestion. The markers have been marked in all Figures.

Q10. Overall, Authors proposed beautiful, meaningful and interesting results, using methods that are not only well defined, but also have a not inconsiderable scientific value. In the discussion, unfortunately, all these points don’t find a correspondence: all the data are not very well argued and, in this way, they lose importance.

Answer: Thanks for reviewer suggestion. I have discussed them in the revised manuscript (Page 9, Lines 249-259).

Q11. Moreover, in the Discussion section, the Authors focused their attention on some criticisms about melatonin dosage. It is an interesting point, but, also in this case, not well delineated and the point is not understandable.

Answer: Thanks for reviewer suggestion. I have moved them to the Material and Method section in the revised manuscript.

Q12. Another very deficient point is the bibliography, which is not adequately updated and that is very thin.

Answer: Thanks for reviewer suggestion. I added several recently published citations.

Q13. Overall, in mine opinion, there are important outcomes obtained through these analyses, but they are not well expressed: this makes the work unclear and the communication of what has been achieved ineffective.

Answer: Thanks for reviewer suggestion. I have discussed them as ” In the present study, a larger adipocyte size in inguinal white adipose tissues was observed in the PC group compared with the VC group. However, melatonin intervention significantly decreased in the large adipocytes compared with the PC group. Additionally, 10 and 20 mg/kg melatonin induced brite/beige adipocytes in iWATs. But brite/beige adipocytes were not discovered in MEL50 may due to the dissected fat area. However, high-dose melatonin supplementation (MEL50) significantly enhanced circulating irisin level compared to PC group. FNDC 5/irisin has recently been identified as a novel protein that can stimulate browning of white fat by inducing thermogenesis through increasing UCP 1 [36]. Melatonin administration also significantly downregulated FNDC5 mRNA and slightly decreased UCP-1 mRNA to restore the irisin resistance status of iWATs. In the current study, we were unable to detect a significant increase of PGC1α expression in melatonin-treated groups. The result is similar to previous study [37] that an additional regulation of FNDC5 independent of PGC1α in adipose tissue. Additionally, melatonin significantly decreased TC level and increased HDL-C level compared to PC group. The hypolipidemic effect of melatonin in our study may have resulted from the increased cholesterol excretion in the feces.” in the revised manuscript.

Q14. Already in the introduction, although the purpose of the work is clear, the reasons for supposing that melatonin can be used as a therapeutic intervention are not clear.

Answer: Thanks for reviewer suggestion. I added some sentences in the introduction of the revised manuscript (Page 2, Lines 50-60).

Q15. In the same way, at the end, the Authors draw conclusions, correct, but without having adequately discussed the results obtained and without having first gutted the mechanisms on which melatonin can work.

Answer: Thanks for reviewer suggestion. I added some discussion in the revised manuscript (Page 9, Lines 248-262).

Q16. Results have been listed but not explained, which makes the results themselves not very useful.

Answer: Thanks for reviewer suggestion. I added some explained in the revised manuscript.

Round 2

Reviewer 2 Report

Authors implemented all my suggestions and substantially improved the manuscript. I recommend the publishing.

Author Response

Thank you for your suggestion regarding our manuscript (molecules-8446471).

We thank the reviewers for the valuable opinions. The manuscript was revised according to your constructive suggestions.

Reviewer 3 Report

Effects of Melatonin on Lipid Metabolism and Circulating Irisin in Sprague-Dawley Rats with Diet-induced Obesity

Abstract, keywords and introduction are well structured.

A single note about one of the keywords that have been changed: I would not use the word “anti-obesity” because it sounds a bit obsolete and not understandable, I think that even just "obesity" could clearly indicates one of the fields of interest of the research.

Perhaps interest could be placed also on the "chronic subministration" to which animals are subjected, I think it is an important point to stress in order to highlights how, the one proposed with the use of melatonin, is not an intervention, which aims at an immediate reversion of the pathological condition, on the contrary, we are talking about a long-term therapeutic strategy that leads to a remodeling of the basic conditions, improving the general ending outcome.

The changes and additions made to the introduction have had the desired effect.

The Authors have added bibliographical references even quite recently, only a few small typing errors and some small grammatical errors remain to be corrected.

The same about the Material and methods section.

In this revised version the results are clearer and better expressed, but there is still no real discussion of the data collected.

It is not clear what the outcome of the study is and what the implications, even if only future, of a possible therapeutic strategy are.

Moreover, considering the limitations that the authors have outlined regarding high dosages of melatonin, the questions remain two:

- does it make sense to report these data?

- personally I think that it is scientifically useful to share this data, also considering this "side effect" of a high dosage of melatonin as an indicators for other future studies. Anyway, it would be useful to bring a bibliography in support of what has been identified, then Authors should report other works that confirm the same difficulties in the administration of high dosages of melatonin.

Again, regarding to the results, considering the data obtained on the expression of PGC1 and UPC, it is not clear at this point how melatonin exerts its action.

The same is true for the changes in lipid profile: the Authors' conclusion, supported by the results, is that the overall improvement in lipid profile is due to increased excretion of cholesterol in the stool, but by what mechanism does this occur?

There is still no interpretative step in the discussion part: the results obtained deserve to be contextualized and evaluated in the light of previous studies.

The work lack explanations of the underlying mechanism that are necessary to answer the following points, which are still open:

- if melatonin acts on fibronectin levels, but then the pathway does not proceed as expected with PGC1 and UPC, how do the Authors explain the recorded effects? Could there be other fibronectin-dependent pathways that still induce an improvement in response to melatonin administration?

- how does melatonin increase faecal cholesterol excretion?

- how does it act, instead, in regulating the levels of circulating irisin? Which pathways does it act on?

Regarding the role of melatonin in regulating WAT and BAT levels and promoting browning, on the other hand, the data are a little better expressed. It is not clear, however, the reason for the discrepancy regarding the presence of beige adipose tissue in the MEL50 group. The Authors talk about a possible not complete concordance due to a cutting error, or a low presence of BAT in that particular segment considered, but there is no different evidence in any of the other animals analyzed?

Could it not be another reason for a different response in these animals?

Finally, I see no change from the first proposed figure in relation to Figure 3. In particular I think it would be important to better identify in the microphotographs the different composition of the analyzed tissue, better labelling with arrows rather than asterisks the portions studied (WAT, BAT or beige adipose tissue).

Author Response

Thank you for your suggestion regarding our manuscript (molecules-8446471).

We thank the reviewers for the valuable opinions. The manuscript was revised according to your constructive suggestions. Attached a file for a point-by-point response to all of the comments.

Round 3

Reviewer 3 Report

I thanks the Author for the work.

Compared to the previous version, the manuscript has undergone an important improvement.

In particular, the advice about the figures, which are now clearer, has been correctly considered.

The results are better expressed and very precise. They represent a real strength of the work.

The discussion is also more articulated and makes the aim and purpose of the whole work clearer.

The only doubt still present is about "the irisin resistance status of iWATs". The role of this pathway is not clear and it would be important to better highlight the impact of re-establishing "the irisin resistance status of iWATs". Which is the impact of this modification on adipose tissue metabolism?

The role of melatonin is now clearer and it is better understood how the study has not focused only on the melatonin-irysin relationship, but how, in general, it considers the role of melatonin in the obesity model.

Once the melatonin-irysin link, which remains somewhat deficient in some places, have been better clarified the work will be complete.

Overall, English has been well improved.

Author Response

Dear,

Thank you for your suggestion regarding our manuscript (molecules-8446471).

We thank the reviewers for the valuable opinions. The manuscript was revised according to your constructive suggestions. The following is a point-by-point response to all of the comments:

For reviewer #3:

Q1. The only doubt still present is about "the irisin resistance status of iWATs". The role of this pathway is not clear and it would be important to better highlight the impact of re-establishing "the irisin resistance status of iWATs". Which is the impact of this modification on adipose tissue metabolism?.

Answer: Thanks for your comment. We have added these sentences in the revised manuscript. As follows:

“FNDC 5/irisin has recently been identified as a novel protein that can stimulate browning of white fat by inducing thermogenesis through increasing UCP 1 (Guilford et al., 2017). Long-term melatonin administration also significantly downregulated FNDC5 mRNA and slightly decreased UCP-1 mRNA to restore the irisin resistance status of iWATs. Therefore, melatonin increased circulating irisin, resulting from the cleavage of the membrane protein FNDC5, to induce adipocyte browning, with increased lipid oxidation and thermogenesis. So, melatonin induced WAT browning that is a promising means to increase energy expenditure and improve lipid metabolism.” (Page 9, Lines 261-268)

Reference

Guilford, B.L.; Parson, J.C.; Grote, C.W.; Vick, S.N.; Ryals, J.M.; Wright, D.E. Increased FNDC5 is associated with insulin resistance in high fat-fed mice. Physiol. Rep. 2017, 5, e13319.

Q2. The role of melatonin is now clearer and it is better understood how the study has not focused only on the melatonin-irysin relationship, but how, in general, it considers the role of melatonin in the obesity model.

Answer: Thanks for reviewer’s reminder. We have added these sentences in the revised manuscript. As follows:

“In addition, high-dose melatonin supplementation (MEL50) significantly enhanced circulating irisin level compared to PC group. In the study, we found that melatonin could increase circulating irisin level, which binds to the surface of white adipocytes to induce the expression of UCP­1 and trigger the transformation of white adipocytes into brown adipocytes. Browning of adipose tissue causes dissipation of energy in the form of heat without ATP formation that may enhance fatty acid oxidation and increase energy expenditure through non shivering thermogenesis. Therefore, melatonin inhibites the occurrence and development of obesity through enhancing circulating irisin level.” (Page 9, Lines 254-261)